

# Digital security vulnerabilities and threats implications for financial institutions deploying digital technology platforms and application: FMEA and FTOPSIS analysis

Abeeku Sam Edu[1], Mary Agoyi[2] and Divine Agozie[1]

[1] Management Information Systems, Cyprus International University, Nicosai, Cyprus
[2] Information Technology, Cyprus International University, Nicosia, Cyprus

## ABSTRACT

Digital disruptions have led to the integration of applications, platforms, and infrastructure. They assist in business operations, promoting open digital collaborations, and perhaps even the integration of the Internet of Things (IoTs), Big Data Analytics, and Cloud Computing to support data sourcing, data analytics, and storage synchronously on a single platform. Notwithstanding the benefits derived from digital technology integration (including IoTs, Big Data Analytics, and Cloud Computing), digital vulnerabilities and threats have become a more significant concern for users. We addressed these challenges from an information systems perspective and have noted that more research is needed identifying potential vulnerabilities and threats affecting the integration of IoTs, BDA and CC for data management. We conducted a step-by-step analysis of the potential vulnerabilities and threats affecting the integration of IoTs, Big Data Analytics, and Cloud Computing for data management. We combined multi-dimensional analysis, Failure Mode Effect Analysis, and Fuzzy Technique for Order of Preference by Similarity for Ideal Solution to evaluate and rank the potential vulnerabilities and threats. We surveyed 234 security experts from the banking industry with adequate knowledge in IoTs, Big Data Analytics, and Cloud Computing. Based on the closeness of the coefficients, we determined that insufficient use of backup electric generators, firewall protection failures, and no information security audits are high-ranking vulnerabilities and threats affecting integration. This study is an extension of discussions on the integration of digital applications and platforms for data management and the pervasive vulnerabilities and threats arising from that. A detailed review and classification of these threats and vulnerabilities are vital for sustaining businesses' digital integration.

Corresponding author
Abeeku Sam Edu, asedu@ug.edu.gh

## INTRODUCTION

Emerging and ubiquitous digital applications have created opportunities for industries to combine technologies to improve operations through open digital interoperability, scalability, and interdependence for digital platforms and applications collaboration (*Kebande, Karie & Venter, 2017*). Digital open collaboration has made it possible for applications, tools, and platforms to merge or synchronize successfully with other applications. Recent studies have identified that emerging digital applications such as the Internet of Things (IoTs), Big Data Analytics (BDA), and Cloud Computing (CC) can synchronize to support data sourcing, data analytics, and storage on a single platform (*Stergiou et al., 2018*). For example, IoT applications such as radio-frequency identification devices (RFID) and other actuators are primarily used to source data from different fields to support BDA for data processing and insights (*Yang et al., 2017*). CC provides access to shared resources which enable IoTs data collection for real-time data analysis (*Atlam et al., 2018*). The concept of digital disruptions has led firms to integrate digital applications and platforms capabilities to promote business values.

Accordingly, financial service operations have been largely influenced by digital applications and platforms to build innovative services and ultimately to increase revenue. For most banks, the need to improve data sourcing and insight creation from data plus their ability to store large volumes of customer data has led to the adoption of IoTs, BDA and CC for financial service operations. *Feher & Varga (2017)* posited that "the changing role of branches, mobile and phone-based services and products and services" have also contributed to the drive towards ubiquitous platforms and applications. Specifically, adopting the IoTs as the "Bank of Things" has helped commercial banks use automated teller machine kiosks to directly interact with customers' mobile phones to easily withdraw money without using a debit or credit card. IoTs connected devices are valuable for transmitting customers' financial transactions. This in turn allows financial institutions to collect, exchange, and create insight from each transatction. According to the *Cybersecurity Observatory Finder (2020)*, "Bank of Things (BoTs) is the material infrastructure that facilitates the billions of data transfers that take place every day". Most banks have also acknowledged that the amount of data being generated has increased enormously due to different sources of collecting data. As such, data have become the most vital asset for banks to effect changes for financial services operations. The focus for banks is the ability to create value, insight and leverage from data assets. Most banks have therefore construed big data into "a greater scope of information, new kinds of data and analysis, real-time information, data influx from new technologies, modern media, large volumes of data, the latest buss word and data from social media" (*Forest et al., 2014*). New digital technologies have further classified data into volumes, variety, velocity, veracity and value. Personal data and data from daily financial transactions have been optimized using big data analytical tools to create new financial business models, collaborations among employees, fraud detection, optimizing financial operations and customer-focused services. Commercial and retail banks use big data analytics tools such as data mining, query and reporting, data visualization tools, and streaming analytics, to analyze data for specific business models and

operational improvements. Cloud computing infrastructures are used by approximately 89% of banks globally as of 2015 (*Hon & Millard, 2018*). Although most banks were initially hesitant to transfer core data to the cloud, the deployment of cloud services is now accepted by commercial banks to support operations. The use of cloud computing services can provide continuous banking services across branches and integrate customer data or information in all branches. Virtual cloud computing services have been used to support the IoTs and BDA, allowing them to have digital scalability, collaborations, interoperability, interdependence, and data management processes within a digital ecosystem.

Despite the benefits of integrating digital applications for collaboration, scalability and cost-efficiency, these applications are complex (*Yang et al., 2017*; *Heavin & Power, 2018*). Studies have indicated that insufficient standardization, heterogeneity, Internet availability, and infrastructure limit the success of digital integration for data management (*Kache & Seuring, 2015*). At the heart of these complexities are digital security risks and vulnerabilities. Digital ecosystem interactivities are perpetually affected by threats and vulnerabilities as a result of network connectivity for data transmission and storage via the Internet (*Manogaran et al., 2018*). Consequently, firms relying on Internet accessibility for digital platforms, and applications interactivity for data management constantly deal with digital security threats and vulnerabilities. The dependence on emerging digital innovations leaves businesses prone to more digital security attacks. These risks stem from the combination of threats within the digital environment. Studies have suggested that digital security risks and vulnerabilities are a result of threats from digital platform usage, the physical environment, people, and an organization's digital ecosystem (*OECD, 2015*). These dangers affect data integrity, confidentiality, and availability, preventing them from integrating successfully into emerging digital platforms.

Attempts made by IoTs, BDA, and CC to address digital security attacks, digital resources, and the environment have minimized these effects in a number of ways (*Yan et al., 2020*; *Xu et al., 2020*; *Xu et al., 2019*). A cursory review of these approaches in information system (IS) research leads to either technical or managerial perspectives of related security attacks and vulnerabilities (*Flores, Antonsen & Ekstedt, 2014*; *Singh, Gupta & Ojha, 2014*; *Joshi & Singh, 2017*). Again, attempts to address digital security vulnerabilities presented by these applications have been treated independently specific to platforms or application deployment (*Sicari et al., 2015*; *Chang, Kuo & Ramachandran, 2016*). There is little research on the implications of potential vulnerabilities and threats for digital technology integration of IoTs, BDA, and CC in data management (*Cherdantseva et al., 2016*). There is also an insufficient understanding of how to address risk when integrating these three applications (*Choo et al., 2018*). We sought to understand the potential vulnerabilities and threats arising from integrating IoTs, BDA, and CC applications to provide security managers with a better awareness of threats against digital interdependence on a single platform.

Assessing digital security threats and vulnerabilities requires continuous efforts to identify, analyze, and measure the attacks with appropriate security management techniques. *Bojanc & Jerman-blaz (2008)* suggested that the attempt to assess the impact of digital security risks and vulnerabilities should include identifying and assessing loss caused by successful attacks. It should also include decisions to mitigate or reduce the

operational risk. Similarly, *Chen & Zhao (2013)* advocated that security risk assessment must broadly identify the security environment and the accompanying risks issues. Steps must be taken to ensure comprehensive analysis, measurement, and control of the potential risk failures for digital technology resources. Implementing these steps incorporates the probability of identifying risks to the digital systems, detecting the extent of the impact, the severity of potential incidents, procedures for minimizing security controls, and monitoring approved controls' efficacy (*Silva et al., 2014*; *Munodawafa & Awad, 2018*). We examined the literature and identified twenty-seven vulnerabilities and threats that may affect IoTs, BDA, and CC integration. Vulnerabilities and threats were further grouped under access control vulnerabilities, network security attacks, data and information management, infrastructure attacks, security management failures, identity management, and communication security (*Li & Tang, 2013*; *Kebande, Karie & Venter, 2017*; *Ouaddah et al., 2017*; *Kumar, Raj & Jelciana, 2018*; *Chang, Kuo & Ramachandran, 2016*). We sought to provide a step-by-step analysis of potential vulnerabilities and threats using multi-dimensional analysis such as Failure Mode Effect Analysis (FMEA) and Fuzzy Technique for Order of Preference by Similarity for Ideal Solution (FTOPSIS). FMEA and FTOPSSIS were used to categorize and prioritize the potential vulnerabilities and threats in IoTs, BDA, and CC integration. Our objectives were to:

1. Investigate potential vulnerabilities and threats affecting the integration of IoTs, BDA, and CC for data management;
2. Evaluate the potential threats and vulnerabilities using FMEA and Fuzzy TOPSIS;
3. Assess the most prevailing threats and vulnerabilities through risk prioritization ranking.

Our study is structured into six main sections. 'Related Work'' reviews potential digital security threats and vulnerability dimensions for IoTs, BDA, and CC integration. 'FMEA and Fuzzy theory application' presents risk management assessment tools, FMEA, and Fuzzy TOPSIS. 'Methodology' introduces the methodology. 'Analysis and results' presents the analysis of our results, and 'Discussion' and 'Conclusions' are a discussion of the results and the conclusion of the study, respectively.

## RELATED WORK

### Digital security consideration in IoTs, CC and BD integration

Studies have attempted to classify digital security threats and vulnerabilities for IoTs, CC, and BDA into security risk dimensions. A cursory review of the literature found that the risks influenced the overall benefits of deploying IoTs, CC, and BDA for data management. According to *Li & Tang (2013)*, the identification and prioritization of critical threats and vulnerabilities should find security dimensions to be potential threats to digital platforms and the use of applications. Our study considered these threats when classifying the vulnerabilities and risks for the deployment and use of IoTs, CC and BDA.

### *Infrastructure (INF) vulnerabilities and attacks*

Digital infrastructure disruptions affect built-in systems in the digital environment. Infrastructure vulnerabilities and risks cause disruption or impact occurrences; they

affect hardware and network resources for digital platform interactivity (*Li & Tang, 2013*). Vulnerabilities in the infrastructure include IT automated systems failure through hardware malfunctions, natural disasters, or loss of electric power (*Xu & Masys, 2016*). Integrating IoTs, BDA, and CC infrastructures support complex data structures; however, they are targets for hackers (*Kebande, Karie & Venter, 2017*).

Reliance on such digital platform interactivity depends on the security of a pool of shared physical-digital resources. Failures eventually disrupt the interdependence of software platforms that facilitate interoperability connectivity among IoTs, CC, and BDA (*Chatzipoulidis, Michalopoulos & Mavridis, 2015*). Studies have shown that the heterogeneity of digital devices or resources result in a higher likelihood of digital infrastructure failures on a platform that supports interoperability (*Ullah et al., 2017*). Digital infrastructure failures are further attributed to disruption due to a lack of back-up power, poor patch updates, and the use of infrastructure (*Cobb et al., 2018*).

### Security management (SM) failures

Security management failures occur due to the inadequate use of digital security safety measures, insufficient security audits, and poor maintenance of hardware and software assets (*Soomro, Shah & Ahmed, 2016*). Failing to adopt a holistic approach to the daily management of security occurrences often disrupts security with digital resources. Poor security measures towards cloud services such as infrastructure as a service (IaaS) in the layers may affect the delivery of services to either a third party or an organization using IoTs and BDA for data management (*Jouini & Rabai, 2017*). Security management failures are also attributed to a lack of system security audits, security policy review, and hardware and digital resource maintenance (*Le, Hartog den & Zannone, 2018*).

### Communication security (CS) failures

A review by *Bays et al. (2015)* suggested that communication security failures and threats with a lack of security encryption protocols primarily affected digital platform communication. Lack of communication security requirements on IoTs, CC and BDA platforms affected data and information integrity, confidentiality and authentication (*Martin et al., 2017*). The interaction of communication platforms through data and information sharing were compromised, affecting data privacy, integrity and confidentiality. Communication channels were further compromised on a wireless network or interface facilitating the integration of IoTs, CC and BDA (*Shu et al., 2016*).

### Identity management (IDM) failures

Identity management secures the identification and notification of users' activities on digital platforms and resources. Identity management ensures unique and standardized identification that virtually authenticates users on a secure platform to ensure their safety and security (*Ferreira & Alonso, 2013*). Identity management security is affected by the reliability and applicability of IDM systems that control CC platform and provide scalability for BDA, and remote access to IoTs actuators for varied connectivity and usage (*Habiba et al., 2014*). *Habiba et al. (2014)* found that identity management security challenges include identity theft, least privileges, elevated privileges, and trust management.

*Indu, Anand & Bhaskar (2018)* suggested that due to outsourcing and third-party management of digital platform interactivity, identity management security failures or vulnerabilities that arise through IoTs, CC, and BDA must be controlled.

### Access control (ACC) failures

Lack of control of third-party activities through cloud computing sourcing on a platform relying on IoTs to transfer data for big data analytics may result in an unsafe transfer of data (*Gharaibeh et al., 2017*). Digital trust issues emanating from access to digital platforms may influence an organization's security strategy. Failure to provide rigorous control measures for authenticating and authorizing users' privileges on a digital platform can complicate IoTs, CC and BDA integration (*Ouaddah et al., 2017*). A systematic analysis of digital security challenges revealed how IoT nodes failed to authenticate authorized access on a cloud platform (*Hossain, Fotouhi & Hasan, 2015*), making it vulnerable to attacks.

### Network security (NS) vulnerabilities

Vulnerability attacks occur through the Internet and system network may affect IoTs, CC, and BDA connectivity. These attacks usually affect physical or virtual networks that facilitate the integration of digital interactivity platforms. *Singh, Jeong & Hyuk (2016)* identified denial of services (DoS), spoofing, distributed denial of service (DDoS), and phishing attacks as network security occurrences affecting access to digital platform integration. These attacks significantly affected IoTs devices that act as a conduit for transmitting data or information through the cloud platform for big data prescriptive analysis (*Hossain, Fotouhi & Hasan, 2015*).

### Data and information management (DINF) vulnerabilities

Key policies must define measures to secure and protect data on digital platforms. Examples of vulnerabilities and threat occurrences affecting data and information management on integrated platforms include lack of data scalability and failure to secure data transferability, failure to provide data privacy, lack of data theft prevention, failure to prevent unauthorized access, and including sensitive information in data storage (*Kumar, Raj & Jelciana, 2018*; *Chang, Kuo & Ramachandran, 2016*). Few studies have explored the benefits obtained through IoTs, CC, and BDA integration; challenges for managing data on these interactive platforms are on the rise (*Cai et al., 2017*).

A summary of security dimensions with accompanying vulnerabilities and threats is shown in Table 1. For each digital security dimension, specific failure modes were highlighted as perceived attacks and failures occurring with IoTs, BDA, and CC deployment. Table 1 shows that access control security vulnerabilities are perceived to include external management failures, failure to manage external and internal media removal, control of third party privileges, and failure to control access to digital platforms. Network security issues also occurred due to failed firewalls, unsuccessful prevention of network attacks, missing intrusion detection and prevention systems and a failure to prevent network attacks. Table 1 shows the remaining vulnerabilities and threats defined under each security dimension.

**Table 1  Failure modes.** Digital security risk and vulnerabilities.

| Dimensions | Failure modes |
| --- | --- |
| Access Control (ACC) | AC1- External management access control |
| | AC2-Management of removable of internal and external media risk |
| | AC3-Control to third party privileges risk |
| | AC4-Access to external digital platforms control |
| Network Security (MS) | NS1- Failure of Firewall protection |
| | NS2- File transfer protocol to authenticate the communication between devices and networks |
| | NS3- Lack of intrusion detection and prevention system |
| | NS4- Lack of preventing network attacks |
| Data and Information Management (DINF) | DINF1- Digital Platforms compatibility failures |
| | DINF2- Reliability of digital platforms |
| | DINF3- Failure of software functionality on platforms |
| | DINF4- Failure of digital configuration with a digital system |
| Infrastructure (INF) | INF1- Failure to control the use of infrastructure |
| | INF2- Lack of infrastructure update and patching |
| | INF3- Software origination and defence failures |
| | INF4- Lack of back-up electric generator |
| Security Management (SM) | SM1- Lack of information security audit |
| | SM2-Lack of a policy paper on digital security safety |
| | SM3- Lack of maintenance of hardware and software |
| | SM4- Lack of security policies reviews |
| Identity Management (IDM) | IDM1- Lack of securing users' true identity |
| | IDM2- Lack of identifying a third-party identity |
| | IDM3- Notification to system administrator on a user's identity |
| | IDM4- Lack of detecting outsourced party activity and identity |
| Communication Security (CS) | CS1- Lack of encryption control management |
| | CS2- Lack of limited content access to internet |
| | CS3- Lack of safety of electronic mail |

### Risk analysis assessment

Risk analysis is the preliminary step in assessing security risk management procedures (*Hinarejos et al., 2018*). Risk analysis is a step-by-step procedure using available information to classify and evaluate different sources of potential risks for the use of digital resources. Its success depends on the ability to correctly identify countermeasures to mitigate risks. According to *Bojanc & Jerman-blaz (2008)*, risk management analysis requires the verification of the likelihood that a risk will occur, the likelihood of detecting the risk, and the consequential effect of the risk should it occur. Risk assessment methods must identify the occurrence of vulnerabilities and threats, evaluate and measure their impact, and detect present and future attacks. We reviewed this multi-dimensional methodology using FMEA and Fuzzy TOPSIS. The two techniques informed the basis of this study and helped to

identify the occurrence of risks and vulnerabilities, the severities of such vulnerabilities to digital applications, and the detectability of continuous implications of such security dimensions.

## FMEA AND FUZZY THEORY APPLICATION

FMEA has been shown to be a useful analytical tool for evaluating potential risk identification failures and preventative measures. FMEA is defined by *Stamatis (2003)* as "an analysis technique for defining, identifying and eliminating known or potential failures, problems, errors and so on from system, design, process and services before they reach the customer". FMEA outlines the process of identifying potential failure modes, causes, effects, and challenges affecting the overall systems, hardware reliability, software applications, and the safety of the system (*Kim & Zuo, 2018*). FMEA identifies the potential failure modes based on their criticality to the systems (*Kangavari et al., 2015*). Measuring the risk priority number (RPN) is determined as the product of Occurrence (Occ), Severity (Sev), and Detection (Det) of a failure mode defined in Eq. (1). In Eq. (1), Occ is the frequency of occurrence of the failure mode, Sev is the extent of the effect of the failure mode, and Det is the probability of detecting the failure before it impacts each system. Groups of decision-makers evaluate the three risk parameters (Occ, Sev, and Det) by providing an assessment value with specific scales for each identified failure mode. A high RPN for any failure mode requires adequate attention to provide corrective measures to the system.

$$RPN = Occ \times Sev \times Det \tag{1}$$

The use of FMEA has been combined with other techniques to improve its efficacy (*Liu, Liu & Liu, 2013*). *Zadeh (1965)* developed the fuzzy set theory to address phenomena characterized by uncertainty or complexities under FMEA conditions. The fuzzy set is able to offer more accurate results with the subjective opinions of FMEA experts. *Hadi-Venchec & Aghajani (2013)* proposed a fuzzy analysis to examine expert views using linguistic terms to evaluate their independent judgment to control failures. Likewise, *Carpitella et al. (2018)* proposed a combined multi-criteria approach to support FMEA for a group of experts to optimize maintenance activities.

Fuzzy sets are expressed in linguistic terms through fuzzy triangular or trapezoidal numbers (*Ramzali, Reza & Ghodousi, 2015*). Linguistic variables represent triangular or trapezoidal fuzzy numbers quantitatively to reflect the responses given by experts (*Zadeh, 1965*). The linguistic variables are expressed to show the fuzzy ratings for failure modes to determine the weighted criteria of risk factors. The linguistic terms and fuzzy numbers for Occ indicate the probability of a failure mode occurring. Severity explains the level of impact of the failure mode affecting the system. The Det scale also shows the extent to which the system could identify failures modes within a specified period.

Crisp RPN values have been criticized due to the subjectivity of quantifying the linguistic scale although FMEA risk analysis has yielded several results. Focus on fuzzy number aggregation to determine the ranking of RPN for risk factors has been criticized because

it does not reflect a fair representation of FMEA group assessments. In response to these limitations, approaches such as a technique for ordering preference by similarity to ideal solution (TOPSIS), analytic hierarchy process (AHP), and data envelopment analysis (DEA) have been proposed in the literature. The TOPSIS and AHP techniques seek to support decision-makers with alternatives under certain conditions (*Sun, Wu & Liu, 2006*). In line with the above, we adopted the FTOPSIS multi-criteria decision method to support FMEA in estimating potential vulnerabilities and threats. The FTOPSIS extends traditional TOPSIS to improve the application of linguistic variables for rating criteria for failure modes under FMEA and fuzzy environment (*Chen, 2000*).

## METHODOLOGY

We present the research design, the sampling method, questionnaire design, data collection and analysis in the following sections to provide holistic insight into the prevailing threats and vulnerabilities in IoTs, BDA, and CC integration for data management.

### Research design

We adopted a mixed methodology approach combining qualitative and quantitative methodologies (*Creswell, 2014*). A qualitative method was used to identify and explore the relevant literature detailing different security threats and vulnerabilities affecting the integration of IoTs, CC, and BDA. The survey method was used to administer the questionnaire for data collection. We used FMEA and Fuzzy TOPSIS techniques as the overarching methodology to evaluate, measure, and prioritize vulnerabilities and risk to achieve the objectives of the research. The application of these techniques bridged the qualitative and quantitative analysis.

### Sample and sampling technique

Security vulnerabilities and threat occurrences requires respondents to have the technical abilities to manage digital risks and an understanding of their impact on emerging digital platforms and applications. We used the purposive non-probability sampling technique to select experts and collect data. The purposive sampling technique was chosen because of its ability to support the responses from individual respondents. Experts were selected for their knowledge of IoTs, CC, and BDA use in Ghanaian financial institutions. The institutions represented international and domestic financial banks. Digital technologies have supported the financial sector over the past five years and the central bank has been instrumental in supporting the financial sectors with the digitization of banking operations due to legislation (*Opoku-Afari, 2019*). Therefore, commercial banks in Ghana are using different digital platforms and applications to tailor financial services to improve processes and customer satisfaction. Ghanaian banks were used to investigate digital security vulnerabilities and threats for digital platforms and application deployment within the financial sector.

### Questionnaire design and data collection

Our questionnaire was developed based on *Goodman*'s (*1996*) and the empirical applications of FMEA made by *Lin et al. (2014)* and *Liu et al. (2012)*. We used a 1–10

linguistic scale (absolutely little influence = 1–2-points and very high influence = 9–10 points) to evaluate, measure, and prioritize security vulnerabilities affecting IoTs, BDA, and CC using Occ, Sev, and Det risk parameters. The parameters represent a group decision matrix corresponding to trapezoidal fuzzy numbers constructed to aggregate expert ratings. The linguistic scales were further converted into trapezoidal fuzzy numbers for Occ, Sev, and Det for each decision maker's response. The trapezoidal fuzzy numbers for each decision maker's rating were set within the [0,1] range were 1–2 = 0; 0; 0.15; 0.2, 3–4 = 0.15; 0.2; 0.35; 0.4, 5–6 = 0.35; 0.4; 0.55; 0.6, 7–8 = 0.55; 0.6; 0.75; 0.8 and 9–10 = 0.75; 0.8; 0.9;1.

The questionnaires were administered to digital security experts within the financial sector in Ghana, a middle-income Sub-Saharan African country. Data were collected after an initial assessment of security experts to ascertain their knowledge of IoTs, CC, and BDA deployment and usage. A total of 315 questionnaires were distributed to 23 financial institutions. A total of 255 responses were obtained, of which 234 were considered suitable for the analysis. The remaining 21 questionnaires were eliminated from our analysis due to incomplete responses.

## Implementation of data analysis

The FMEA and Fuzzy TOPSIS analysis involved data aggregation for a fuzzy group and weight matrix, a fuzzy normalized matrix, a fuzzy ideal solution, and calculation of coefficient scores closeness. We used FMEA to categorize threats and vulnerabilities by the probability of occurrence, the severity of occurrence, and the extent to which the vulnerabilities were detected. Fuzzy TOPSIS further defined the fuzzy set functions. This enabled us to aggregate experts' responses regarding the occurrence, severity, and detection of digital threats and vulnerabilities. This in turn allowed us to determine the group decision matrix. Considering the subjective nature of expert opinions, the Fuzzy TOPSIS also provided a systematic step to normalize the aggregated responses before determining the criticality of values to rank and prioritize the failure modes under investigation.

The experimental design for the methodology is presented in Fig. 1, detailing a step-by-step procedure used to achieve the research objectives. The experimental design provided a statistical procedure for data collection and analysis to yield valid and objective conclusions for this study (*Montgomery, 2017*). Our experimental design began by identifying experts and collecting data. Experts were identified based on their understanding of the terms and specific data for each digital security criterion. Once this stage was satisfied, the questionnaires were distributed to the appropriate experts within the IT security units. The data reflected the extent to which risks and vulnerabilities occurred and their severity and detectability of the use of IoTs, BDA, and CCs. The second stage of our study transformed the linguistic scales of expert rankings into associated fuzzy trapezoidal numbers using Excel Visual Basic to ensure the reliability of the data set for mathematical modelling. Thus, for the fuzzification of stages 3 to 6 we used Fuzzy TOPSIS mathematical modelling to derive the aggregated fuzzy group matrix, the weighted fuzzy matrix, the normalized fuzzy matrix, the distance for ideal solutions, and the closeness of coefficient scores. We

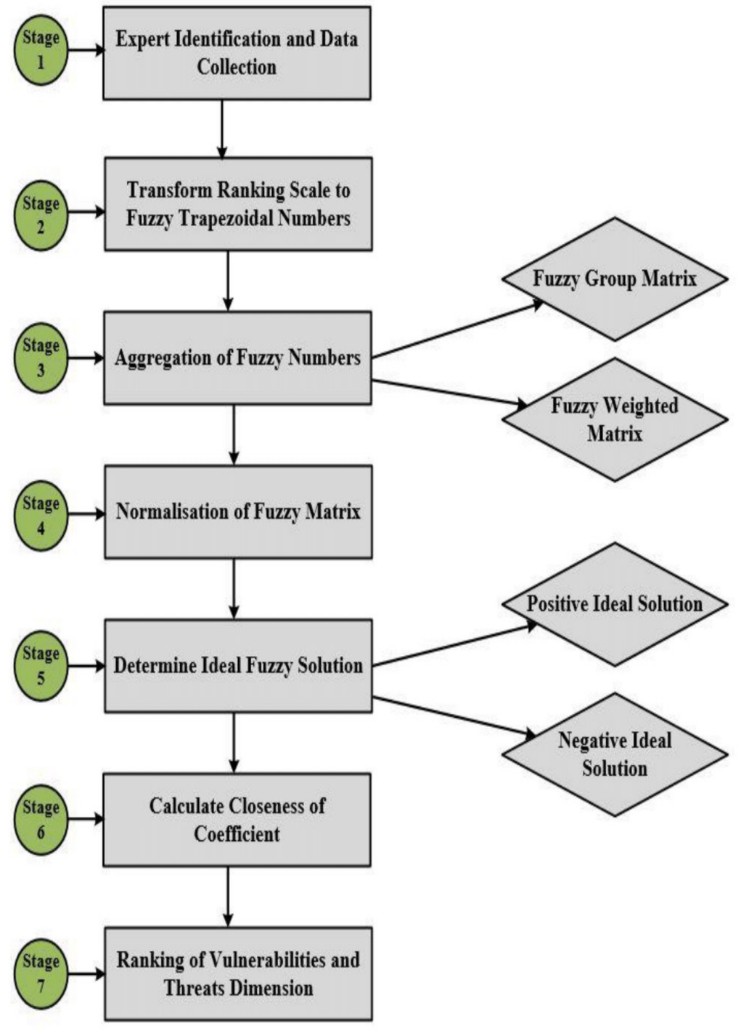

**Figure 1** **Flow chart of the digital security assessment through FMEA and FTOPSIS.**

used the closeness of coefficient scores to prioritize and rank the vulnerabilities and threats identified in the context of digital security risks.

## ANALYSIS AND RESULTS

### Step 1

Fuzzy set A is defined as the membership function that includes elements of the universe X to the unit interval [0,1] (*Zadeh, 1965*). Accordingly, fuzzy set A in X is characterized by membership function $f_A(x)$, the corresponding points for each X must be real numbers in the interval [0,1] representing x with set A (*Zadeh, 1965*). The value of $f_A(x)$ is assumed to be more significant to the membership of X to set A if it is closer to 1. We adapted the trapezoidal fuzzy number $\tilde{A}$ which is represented as $(a_1, a_2, a_3, a_4)$ as shown by *Ramzali,*

*Reza & Ghodousi (2015)* in Eq. (2).

$$f_A(x) = \begin{cases} 0, & \text{if } x < a_1 \\ \dfrac{x-a_1}{a_2-a_1}, & \text{if } a_1 \leq x \leq a_2 \\ 1, & \text{if } a_2 \leq x \leq a_3 \\ \dfrac{x-a_4}{a_3-a_4}, & \text{if } a_3 \leq x \leq a_4 \\ 0, & \text{if } x < a_4 \end{cases} \tag{2}$$

For any given two positive trapezoidal numbers, $\tilde{A} = (a_1, a_2, a_3, a_4)$ and $\tilde{B} = (b_1, b_2, b_3, b_4)$ with positive real numbers, the basic operations for fuzzy set theory are defined in Eqs. (3) to (6) (*Zadeh, 1965*; *Bojadziev & Bojadziev, 2007*).

$$\tilde{A} \oplus \tilde{B} = [a_1 + b_1, a_2 + b_2, a_3 + b_3, a_4 + b_4] \tag{3}$$

$$\tilde{A} \ominus \tilde{B} = [a_1 - b_1, a_2 - b_2, a_3 - b_3, a_4 - b_4] \tag{4}$$

$$\tilde{A} \otimes \tilde{B} = [a_1 b_1, a_2 b_2, a_3 b_3, a_4 b_4] \tag{5}$$

$$\tilde{A} \otimes r = [a_1 r, a_2 r, a_3 r, a_4 r] \tag{6}$$

Again, given $m$ as cross-functional decision makers $DM_k(, 2, \ldots, m) k = 1$ in a FMEA team is responsible for evaluating a set of $n$ failure modes $FM_i$ ($i = 1, 2, \ldots, n$) with respect to Occurrence, Severity, and Detection risk factors. Therefore, $a_{ij1}^k, b_{ij2}^k, c_{ij3}^k, d_{ij4}^k$ are the fuzzy ratings provided by each $DM_k$ to evaluate $FM_i$ for Occ, Sev, and Det expressed in Eqs. (7), (8) and (9).

$$Occ = \left( a_{ij1}^k, b_{ij2}^k, c_{ij3}^k, d_{ij4}^k \right) \tag{7}$$

$$Sev = \left( a_{ij1}^k, b_{ij2}^k, c_{ij3}^k, d_{ij4}^k \right) \tag{8}$$

$$Det = \left( a_{ij1}^k, b_{ij2}^k, c_{ij3}^k, d_{ij4}^k \right) \tag{9}$$

The aggregated fuzzy group decision matrix was derived from *Ghoushchi, Yousefi & Khazaeili (2019)* as formulated in Eq. (10). Given that $\widetilde{FM}_I$ = DM assessment of $\widetilde{FM}_I$ ($i = 1, 2, \ldots, m$) with respect to Occ, Sev, and Det risk factors. The trapezoidal values for Occ, Sev, and Det are calculated for each failure mode under their respective dimensions.

$$\widetilde{FM}_i = \left( \tilde{A}_{ij} = \frac{1}{k} * \sum_k^l a_{ij1}^k, \tilde{B}_{ij} = \frac{1}{k} * \sum_k^l b_{ij2}^k, \tilde{C}_{ij} = \frac{1}{k} * \sum_k^l c_{ij3}^k, \tilde{D}_{ij} = \frac{1}{k} * \sum_k^l d_{ij4}^k \right). \tag{10}$$

Similarly, each aggregated fuzzy weight denoted as $\tilde{w}_j$ was derived using Eq. (11) with respect to Occ, Sev, and Det factors for each failure mode by each $DM_k$.

$$\tilde{w}_j = (w_{j1}, w_{j2}, w_{j3}, w_{j4}) = \left( \tilde{w}_1 = \frac{w_{j1}}{k}, \tilde{w}_2 = \frac{w_{j2}}{k}, \tilde{w}_3 = \frac{w_{j3}}{k}, \tilde{w}_4 \frac{w_{j4}}{k} \right) \tag{11}$$

## Step 2

The cost attributes procedure was used to compute the $\tilde{R}_{ij}$ in Eq. (12) to determine the normalized decision matrix since vulnerabilities and risk are used for this study. The cost attribute in Eq. (13) represents the minimum value for $a_j^-$ for each vulnerability respective to Occ, Sev, and Det. Therefore, the normalized fuzzy weights ($\tilde{z}_{ij}$) were computed using Eq. (16) for the group decision matrix for each vulnerability and risk failure mode for Occ, Sev, and Det; the result is in Table 2.

$$N = \begin{bmatrix} r_{11} & r_{12} & \cdots & r_{1j} \\ r_{21} & r_{22} & \cdots & r_{2j} \\ \vdots & \cdots & \ddots & \vdots \\ r_{i1} & r_{i2} & \cdots & r_{ij} \end{bmatrix} \tag{12}$$

$$\tilde{r}_{ij} = \begin{cases} \left( \dfrac{a_{ij}}{d^+}, \dfrac{b_{ij}}{d^+}, \dfrac{c_{ij}}{d^+}, \dfrac{d_{ij}}{d^+} \right) & \text{if } j \text{ is a benefit attribute} \\[3ex] \left( \dfrac{a_j^-}{d_{ij}}, \dfrac{a_j^-}{c_{ij}}, \dfrac{a_j^-}{b_{ij}}, \dfrac{a_j^-}{a_{ij}} \right) & \text{if } j \text{ is a cost attribute} \end{cases} \tag{13}$$

$$d_j^+ = \max \ of \ d_{ij} \ \text{if } j \text{ is a benefit attribute} \tag{14}$$

$$a_j^- = \min \ of \ a_{ij} \ \text{if } j \text{ is a cost attribute} \tag{15}$$

$$\tilde{z}_{ij} = w_j . \tilde{r}_{ij} \tag{16}$$

Where $\tilde{r}_{ij} = \left( \frac{a_j^-}{d_{ij}}, \frac{a_j^-}{c_{ij}}, \frac{a_j^-}{b_{ij}}, \frac{a_j^-}{a_{ij}} \right)$ as described in Eq. (13).

## Step 3

In the next stage we computed the fuzzy ideal positive and negative solution in Eqs. (17) to (23) (*Carpitella et al., 2018*). The Fuzzy Positive Ideal Solution (FPIS) and Fuzzy Negative Ideal Solution (FNIS) are further defined in Eqs. (17) and (18), respectively (*Chen, 2000*).

$$V^* = (\tilde{z}_{i1}^*, \tilde{z}_{i2}^*, \dots \tilde{z}_{ij}^*) \tag{17}$$

$$V^- = (\tilde{z}_{i1}^-, \tilde{z}_{i2}^-, \dots \tilde{z}_{ij}^-) \tag{18}$$

Where

$$\tilde{z}_{ij}^* = \max \tilde{v}_{ij}(\max a_{ij}, \max b_{ij}, \max c_{ij}, \max d_{ij}) \tag{19}$$

$$\tilde{z}_{ij}^- = \min \tilde{v}_{ij}(\min a_{ij}, \min b_{ij}, \min c_{ij}, \min d_{ij}) \tag{20}$$

The distance for two generic Trapezoidal Fuzzy Numbers (TrFNs) is calculated in Eq. (21). For the crisp value for each failure mode, this was done after determining $\min \tilde{v}_{ij}$ and $\max \tilde{v}_{ij}$ for Occ, Sev, and Det (*Khalili-damghani & Sadi-nezhad, 2013*). Therefore,

**Table 2  Normalized group fuzzy weighted matrix.** Group weighted matrix.

| Dimensions | Failure mode | Occurrence | | | | Severity | | | | Detection | | | |
|---|---|---|---|---|---|---|---|---|---|---|---|---|---|
| Access Control | AC1 | 0.3025 | 0.3665 | 0.5945 | 0.7150 | 0.3385 | 0.4094 | 0.6327 | 0.7550 | 0.2126 | 0.2649 | 0.4863 | 0.5970 |
| | AC2 | 0.3027 | 0.3667 | 0.5946 | 0.7150 | 0.3385 | 0.4082 | 0.6327 | 0.7550 | 0.2086 | 0.2606 | 0.4828 | 0.5970 |
| | AC3 | 0.3036 | 0.3676 | 0.5951 | 0.7150 | 0.3353 | 0.4018 | 0.6330 | 0.7550 | 0.2100 | 0.2607 | 0.4837 | 0.5970 |
| | AC4 | 0.3010 | 0.3639 | 0.5937 | 0.7150 | 0.3389 | 0.4078 | 0.6348 | 0.7550 | 0.2109 | 0.2621 | 0.4843 | 0.5970 |
| Network Security | NS1 | 0.3059 | 0.3730 | 0.6089 | 0.7150 | 0.3424 | 0.4139 | 0.6389 | 0.7550 | 0.2129 | 0.2658 | 0.4870 | 0.5970 |
| | NS2 | 0.2978 | 0.3597 | 0.5930 | 0.7150 | 0.3378 | 0.4083 | 0.6333 | 0.7550 | 0.2086 | 0.2603 | 0.4822 | 0.5970 |
| | NS3 | 0.3070 | 0.3733 | 0.5968 | 0.7150 | 0.3353 | 0.4035 | 0.6321 | 0.7550 | 0.2057 | 0.2566 | 0.4816 | 0.5970 |
| | NS4 | 0.3059 | 0.3699 | 0.5962 | 0.7150 | 0.3398 | 0.4104 | 0.6352 | 0.7550 | 0.2097 | 0.2613 | 0.4844 | 0.5970 |
| Data and Information Management | DINF1 | 0.3090 | 0.3752 | 0.5978 | 0.7150 | 0.3341 | 0.4026 | 0.6334 | 0.7550 | 0.2086 | 0.2594 | 0.4848 | 0.5970 |
| | DINF2 | 0.3031 | 0.3661 | 0.5958 | 0.7150 | 0.3283 | 0.3963 | 0.6305 | 0.7550 | 0.1980 | 0.2480 | 0.4779 | 0.5970 |
| | DINF3 | 0.2971 | 0.3594 | 0.5916 | 0.7150 | 0.3364 | 0.4054 | 0.6336 | 0.7550 | 0.2040 | 0.2545 | 0.4818 | 0.5970 |
| | DINF4 | 0.3031 | 0.3673 | 0.5958 | 0.7150 | 0.3313 | 0.3986 | 0.6320 | 0.7550 | 0.2068 | 0.2579 | 0.4826 | 0.5970 |
| Infrastructure | INF1 | 0.3049 | 0.3687 | 0.5977 | 0.7150 | 0.3371 | 0.4063 | 0.6321 | 0.7550 | 0.2094 | 0.2609 | 0.4824 | 0.5970 |
| | INF2 | 0.3048 | 0.3682 | 0.5957 | 0.7150 | 0.3387 | 0.4074 | 0.6338 | 0.7550 | 0.2100 | 0.2611 | 0.4837 | 0.5970 |
| | INF3 | 0.3017 | 0.3641 | 0.5950 | 0.7150 | 0.3389 | 0.4070 | 0.6339 | 0.7550 | 0.2090 | 0.2595 | 0.4821 | 0.5970 |
| | INF4 | 0.3158 | 0.3816 | 0.6104 | 0.7150 | 0.3376 | 0.4057 | 0.6332 | 0.7550 | 0.2143 | 0.2656 | 0.4855 | 0.5970 |
| Security Management | SM1 | 0.3140 | 0.3801 | 0.6003 | 0.7150 | 0.3409 | 0.4113 | 0.6348 | 0.7550 | 0.2097 | 0.2613 | 0.4844 | 0.5970 |
| | SM2 | 0.3107 | 0.3746 | 0.5987 | 0.7150 | 0.3360 | 0.4042 | 0.6324 | 0.7550 | 0.2109 | 0.2619 | 0.4824 | 0.5970 |
| | SM3 | 0.3015 | 0.3632 | 0.5940 | 0.7150 | 0.3360 | 0.4043 | 0.6334 | 0.7550 | 0.2078 | 0.2584 | 0.4833 | 0.5970 |
| | SM4 | 0.3066 | 0.3687 | 0.5957 | 0.7150 | 0.3380 | 0.4063 | 0.6344 | 0.7550 | 0.2093 | 0.2599 | 0.4833 | 0.5970 |
| Identity Management | IDM1 | 0.3077 | 0.3733 | 0.5972 | 0.7150 | 0.3419 | 0.4129 | 0.6354 | 0.7550 | 0.2114 | 0.2630 | 0.4846 | 0.5970 |
| | IDM2 | 0.3036 | 0.3677 | 0.5960 | 0.7150 | 0.3413 | 0.4119 | 0.6360 | 0.7550 | 0.2142 | 0.2665 | 0.4863 | 0.5970 |
| | IDM3 | 0.2998 | 0.3644 | 0.5931 | 0.7150 | 0.3353 | 0.4048 | 0.6330 | 0.7550 | 0.2118 | 0.2635 | 0.4848 | 0.5970 |
| | IDM4 | 0.2959 | 0.3581 | 0.5900 | 0.7150 | 0.3374 | 0.4072 | 0.6360 | 0.7550 | 0.2101 | 0.2618 | 0.4847 | 0.5970 |
| Communication Security | CS1 | 0.3072 | 0.3712 | 0.5969 | 0.7150 | 0.3402 | 0.4095 | 0.6345 | 0.7550 | 0.2152 | 0.2672 | 0.4879 | 0.5970 |
| | CS2 | 0.3004 | 0.3620 | 0.5924 | 0.7150 | 0.3353 | 0.4035 | 0.6321 | 0.7550 | 0.2139 | 0.2654 | 0.4871 | 0.5970 |
| | CS3 | 0.3030 | 0.3656 | 0.5928 | 0.7150 | 0.3346 | 0.4028 | 0.6317 | 0.7550 | 0.2133 | 0.2651 | 0.4858 | 0.5970 |

**Table 3 Fuzzy positive and negative ideal solution.** Distance for FPIS and FNIS.

| Dimensions | Failure mode | Fuzzy Positive Ideal Solution (FPIS) | | | | Fuzzy Negative Ideal Solution (FNIS) | | | |
|---|---|---|---|---|---|---|---|---|---|
| | | Occ | Sev | Det | $d_i^*$ | Occ | Sev | Det | $d_i^-$ |
| | AC1 | 0.0129 | 0.0043 | 0.0019 | 0.0190 | 0.0046 | 0.0084 | 0.0119 | 0.0249 |
| Access | AC2 | 0.0127 | 0.0046 | 0.0053 | 0.0226 | 0.0048 | 0.0079 | 0.0086 | 0.0213 |
| Control | AC3 | 0.0120 | 0.0076 | 0.0046 | 0.0242 | 0.0055 | 0.0046 | 0.0092 | 0.0193 |
| | AC4 | 0.0142 | 0.0041 | 0.0038 | 0.0221 | 0.0032 | 0.0081 | 0.0101 | 0.0213 |
| | NS1 | 0.0066 | 0.0000 | 0.0014 | 0.0080 | 0.0118 | 0.0120 | 0.0124 | 0.0362 |
| Network | NS2 | 0.0166 | 0.0046 | 0.0055 | 0.0267 | 0.0008 | 0.0078 | 0.0084 | 0.0169 |
| Security | NS3 | 0.0091 | 0.0071 | 0.0078 | 0.0240 | 0.0089 | 0.0051 | 0.0060 | 0.0200 |
| | NS4 | 0.0104 | 0.0028 | 0.0044 | 0.0176 | 0.0072 | 0.0094 | 0.0094 | 0.0260 |
| Data | DINF1 | 0.0078 | 0.0075 | 0.0053 | 0.0206 | 0.0103 | 0.0045 | 0.0085 | 0.0234 |
| and | DINF2 | 0.0124 | 0.0120 | 0.0138 | 0.0382 | 0.0050 | 0.0000 | 0.0000 | 0.0050 |
| Information | DINF3 | 0.0173 | 0.0058 | 0.0090 | 0.0321 | 0.0000 | 0.0063 | 0.0048 | 0.0111 |
| Management | DINF4 | 0.0120 | 0.0100 | 0.0068 | 0.0289 | 0.0054 | 0.0020 | 0.0070 | 0.0144 |
| | INF1 | 0.0106 | 0.0057 | 0.0051 | 0.0214 | 0.0068 | 0.0067 | 0.0089 | 0.0224 |
| Infrastructure | INF2 | 0.0114 | 0.0045 | 0.0045 | 0.0204 | 0.0062 | 0.0078 | 0.0093 | 0.0233 |
| | INF3 | 0.0136 | 0.0046 | 0.0057 | 0.0239 | 0.0037 | 0.0077 | 0.0082 | 0.0196 |
| | INF4 | 0.0000 | 0.0055 | 0.0015 | 0.0070 | 0.0173 | 0.0067 | 0.0125 | 0.0366 |
| | SM1 | 0.0052 | 0.0025 | 0.0044 | 0.0121 | 0.0140 | 0.0100 | 0.0094 | 0.0335 |
| Security | SM2 | 0.0073 | 0.0067 | 0.0044 | 0.0183 | 0.0108 | 0.0056 | 0.0097 | 0.0261 |
| Management | SM3 | 0.0142 | 0.0064 | 0.0062 | 0.0268 | 0.0031 | 0.0057 | 0.0076 | 0.0165 |
| | SM4 | 0.0108 | 0.0049 | 0.0052 | 0.0209 | 0.0069 | 0.0072 | 0.0086 | 0.0228 |
| | IDM1 | 0.0088 | 0.0019 | 0.0033 | 0.0139 | 0.0092 | 0.0110 | 0.0106 | 0.0307 |
| Identity | IDM2 | 0.0117 | 0.0019 | 0.0010 | 0.0145 | 0.0057 | 0.0105 | 0.0130 | 0.0292 |
| Management | IDM3 | 0.0146 | 0.0064 | 0.0029 | 0.0239 | 0.0029 | 0.0057 | 0.0109 | 0.0195 |
| | IDM4 | 0.0185 | 0.0044 | 0.0040 | 0.0269 | 0.0012 | 0.0076 | 0.0098 | 0.0186 |
| | CS1 | 0.0095 | 0.0033 | 0.0000 | 0.0128 | 0.0082 | 0.0091 | 0.0138 | 0.0311 |
| Communication | CS2 | 0.0154 | 0.0071 | 0.0012 | 0.0237 | 0.0021 | 0.0051 | 0.0126 | 0.0198 |
| Security | CS3 | 0.0135 | 0.0076 | 0.0017 | 0.0229 | 0.0043 | 0.0046 | 0.0121 | 0.0210 |

$\tilde{A} = (a_1, a_2, a_3, a_4)$ and $\tilde{B} = (b_1, b_2, b_3, b_4)$ represents the TrFNs for each $\widetilde{FM}_i$ for Occ, Sev, and Det factor.

$$d(\tilde{A}, \tilde{B}) \sqrt{\frac{1}{4}\left[(a_1 - b_1)^2 + (a_2 - b_2)^2 + (a_3 - b_3)^2 + (a_4 - b_4)^2\right]} \tag{21}$$

Furthermore, the fuzzy ideal solution for each alternative $d_i$ is then aggregated for the whole set of failure modes for related distances $d^*$ and $d^-$ by Eqs. (22) and (23). Table 3 shows the result of a fuzzy ideal solution for each failure mode.

$$FPIS(d^*) = \sum_{j=1}^{n} d(\tilde{z}_{ij}) \, i = 1, \ldots, n \tag{22}$$

$$FNIS(d^-) = \sum_{j=1}^{n} d(\tilde{z}_{ij}) \, i = 1, \ldots, n \tag{23}$$

### Step 4

Failure mode rankings are finally computed using the closeness of coefficient (CC) in Eq. (24). The final results are shown in Table 4 and indicates the closeness of coefficient score rankings for all vulnerabilities under each dimension (*Javadian et al., 2009*).

$$CC_i = \frac{FNIS}{FNIS + FPIS}. \tag{24}$$

## DISCUSSION

The corresponding fuzzy numbers were calculated using the aggregated group matrix for Occ, Sev, and Det for each vulnerability and threat identified by the experts. The normalized decision matrix was calculated (Table 2) before the final ranking of the vulnerabilities and threats. The results for the final steps are shown in Table 4 and delineate the risk priority ranking for all vulnerabilities and threats using the closeness of coefficient scores. We found:

1. Twenty-seven perceived failure modes constituting threats and vulnerabilities affecting the integration of IoTs, BDA, and CC (Table 1).
2. Using the closeness of coefficient scores, we identified 13 of the 27 vulnerabilities affecting the success of IoTs, BDA, and CC integration (Table 4). The closeness of coefficient scores for these vulnerabilities were all above 0.5. This included failure to control infrastructure (0.511092), lack of security policy review (0.521654), digital platforms compatibility failures (0.531005), lack of infrastructure update and patching (0.532621), external management access control (0.566476), lack of policy paper on digital security safety (0.587907), unsuccessful prevention of network attacks (0.596003), no identification of third-party identity (0.667483), no verification of users' true identity (0.688226), lack of encryption control management (0.70801), failure of firewall protection (0.818448), and lack of backup electric generator (0.838898).

The closeness of coefficient score is indicated by a score closer to or farther from one and implies that vulnerabilities and risk failure modes are ranked from highest to lowest impact. A risk with a high closeness of coefficient is a potential failure significantly affecting the IoTs, BDA, and CCs integration. All vulnerabilities with a closeness of coefficient scores closer to one require more attention during corrective actions to the system. Table 4 and Fig. 2 show "lack of backup electric generator (INF4) under infrastructure risk dimension" with the highest closeness coefficient score of 0.8388 followed by "failure of firewall protection

**Table 4 Failure modes ranking.** Ranking.

| Dimensions | Failure Mode | $d^*$ | $d^-$ | $CC_i$ | Rank |
|---|---|---|---|---|---|
| Access Control | AC1- External management access control | 0.0190 | 0.0249 | 0.566476 | 9th |
| | AC2-Management of removable of internal and external media risk | 0.0226 | 0.0213 | 0.485012 | 15th |
| | AC3-Control to third party privileges risk | 0.0242 | 0.0193 | 0.443452 | 21st |
| | AC4-Access to external digital platforms control | 0.0221 | 0.0213 | 0.491215 | 14th |
| Network Security | NS1- Failure of Firewall protection | 0.0080 | 0.0362 | 0.818448 | 2nd |
| | NS2- File transfer protocol to authenticate the communication between devices and networks | 0.0267 | 0.0169 | 0.387668 | 23rd |
| | NS3- Lack of intrusion detection and prevention system | 0.0240 | 0.0200 | 0.454627 | 17th |
| | NS4- Lack of preventing network attacks | 0.0176 | 0.0260 | 0.596003 | 7th |
| Data and Information Management | DINF1- Digital Platforms compatibility failures | 0.0206 | 0.0234 | 0.531005 | 11th |
| | DINF2- Reliability of digital platforms | 0.0382 | 0.0050 | 0.114851 | 27th |
| | DINF3- Failure of software functionality on platforms | 0.0321 | 0.0111 | 0.257042 | 26th |
| | DINF4- Failure of digital configuration with a digital system | 0.0289 | 0.0144 | 0.332897 | 25th |
| Infrastructure | INF1- Failure to control the use of infrastructure | 0.0214 | 0.0224 | 0.511092 | 13th |
| | INF2- Lack of infrastructure update and patching | 0.0204 | 0.0233 | 0.532621 | 10th |
| | INF3- Software origination and defence failures | 0.0239 | 0.0196 | 0.450525 | 19th |
| | INF4- Lack of back-up electric generator | 0.0070 | 0.0366 | 0.838898 | 1st |
| Security Management | SM1- Lack of information security audit | 0.0121 | 0.0335 | 0.735124 | 3rd |
| | SM2-Lack of a policy paper on digital security safety | 0.0183 | 0.0261 | 0.587907 | 8th |
| | SM3- Lack of maintenance of hardware and software | 0.0268 | 0.0165 | 0.380825 | 24th |
| | SM4- Lack of security policies reviews | 0.0209 | 0.0228 | 0.521654 | 12th |
| Identity Management | IDM1- Lack of securing users' true identity | 0.0139 | 0.0307 | 0.688226 | 5th |
| | IDM2- Lack of identifying a third-party identity | 0.0145 | 0.0292 | 0.667483 | 6th |
| | IDM3- Notification to system administrator on a user's identity | 0.0239 | 0.0195 | 0.449422 | 20th |
| | IDM4- Lack of detecting outsourced party activity and identity | 0.0269 | 0.0186 | 0.408363 | 22nd |
| Communication Security | CS1- Lack of encryption control management | 0.0128 | 0.0311 | 0.70801 | 4th |
| | CS2- Lack of limited content access to internet | 0.0237 | 0.0198 | 0.454612 | 18th |
| | CS3- Lack of safety of electronic mail | 0.0229 | 0.0210 | 0.478928 | 16th |

(NS1) under network security dimension" with a score of 0.818 and "lack of information security audit (SM1) under security management practices" with a score of 0.735. These vulnerabilities and risks potentially disrupt interconnectivity and safety and impact of IoTs, BDA, and CC. The future of digital platforms, applications interactivity, and collaborations depend on addressing associated vulnerabilities such as information security audits and firewall protections (*Mahmoud et al., 2016*). According to *Chatzipoulidis, Michalopoulos & Mavridis (2015)*, digital platforms sustainability depends on the level at which general digital infrastructure is exposed to vulnerabilities. In that regard, *Silva et al. (2014)* reported

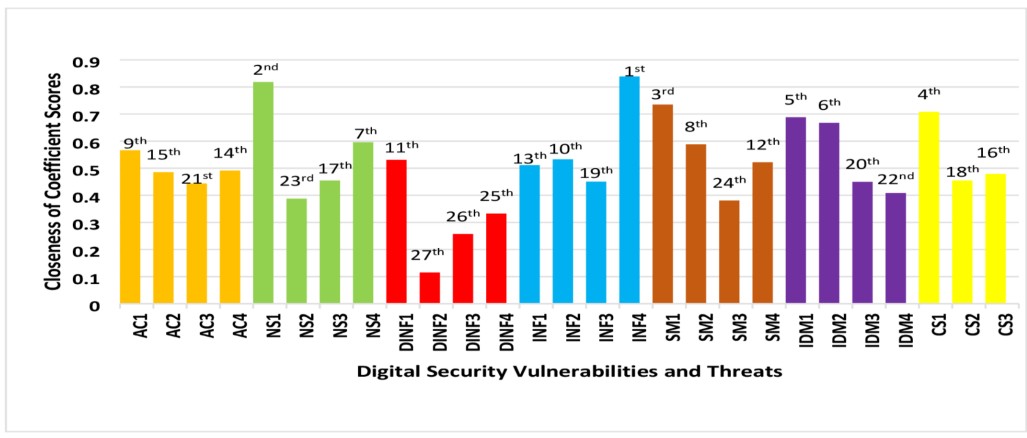

**Figure 2 Closeness of coefficient score and ranking chart.**

that not having a backup electric generator creates vulnerability in an otherwise sustainable digital infrastructure.

Our results suggested that no encryption control management (CS1), not securing a user's true identity (IDM1), and not identifying third parties (IDM2) are the next potential threats associated with communication security and identity management. Notably, the risks associated with communication security was viewed as a challenge to digital platform collaborations (*Soomro, Shah & Ahmed, 2016*; *Silva et al., 2014*). *Bays et al. (2015)* confirmed that failing to ensure security encryption protocols affected digital platform communication. Identity theft and elevated privileges for digital platform use jeopardize the security of identity management. The sustainability of cloud supporting services for BDA and IoTs depends on the effectiveness of identity management security systems within digital platforms interactions (*Habiba et al., 2014*).

Table 4 and Fig. 2 illustrate the remaining vulnerabilities and risks failures with three failure modes under data and information management dimensions; these represent the least ranked threats. Failures of digital configuration with digital systems (DINF4), failures related to software functionality on platforms (DINF3), and reliability of digital platforms (DINF2) were ranked 25th, 26th, and 27th with coefficient scores of 0.332896, 0.257042, and 0.114851, respectively. These were identified as vulnerabilities or failures to digital integration that did not affect IoTs, BDA, and CC deployment. This also suggests an improvement in the integration of IoTs, BDA, and CC to support the use of data and information for decision making (*Ardolino et al., 2018*). In contrast, *Stergiou et al. (2018)* observed an increasing gap in privacy and security issues with data management as a result of integrating IoTs and CC technologies. Applying results above suggests that specific attention should focus on vulnerabilities and risks with the closeness of coefficients scores from 0.8 to 0.5.

## CONCLUSIONS

We sought to provide a comprehensive view of the implications of digital technology integration for both research and practice. We also sought to identify potential digital security threats and vulnerabilities with IoTs, BDA, and CC platform integration to support data management.

Most IS studies have not investigated digital security consequences in IoTs, BDA, and CC using FMEA and FTOPSIS techniques. We provided a review of identifying vulnerabilities and threats likely to affect IoTs, BDA, and CC integration. We also proposed a multi-criteria approach to evaluate the effect of integration holistically. As recognized in this study, research on the vulnerabilities and risks of emerging technologies have been focused on a single platform or application and have independently assessed specific IoTs, BDA, or CC issues (*Sicari et al., 2015*; *Choo et al., 2018*; *Bhathal & Singh, 2019*). Our study provides a holistic theoretical approach towards digital technology integration and its potential impact on digital risk management governance. We offer insight into the various potential vulnerabilities and threats to data management when integrating IoTs applications, BDA, and CC platforms.

The integration of IoTs, BDA and CC capabilities support data source, insight, storage, and knowledge sharing; however, vulnerabilities and threats significantly influence their success. Hence, controlling their vulnerabilities is more critical than focusing only on how they benefit businesses. The results of our study should be used by IT risk managers to assist in identifying vulnerabilities for IoTs, BDA, and CC deployment. The use of FMEA and FTOPSIS alongside other robust digital risk management approaches can be adopted by IT risk managers to support decision-making criteria on the criticality of ranking vulnerabilities to improve information security analysis. IT risk managers should pay greater attention to firewall protection, reliable power and security audit management to reduce recurring attacks on data due to the complexities of IoT, BDA, and CC integration. Our results suggest that internal and external security measures must adequately protect IoT, BDA and CC infrastructure from curtailing frequent attacks from internal and third-party users.

Continuous improvement within the digital ecosystems allows emerging technologies to integrate and promote digital interdependence, interoperability, scalability, and collaboration (*Edu, Agoyi & Agozie, 2020*). It is essential to identify potential complexities accompanying the integration of digital technology applications and platforms. We sought to understand the digital security risks facing the integration of IoTs, BDA, and CC deployment. We identified twenty-seven potential vulnerabilities and threats affecting this process. We used closeness of coefficient scores and found that lack of backup electric generator, firewall protection failure, lack of information security audit, lack of encryption control management, and not securing users' true identity were critical.

Prioritizing vulnerabilities helps with reducing or managing potential digital security risks emanating from digital technology integration. Accordingly, the use of multi-criteria risk management approaches also allows firms and IT security managers to holistically provide corrective actions when digital security fails.

Our findings highlight digital security risk management implications for IoTs, BDA, and CC integration.

Our study is limited by its reliance on security experts from financial institutions as their views may not reflect the views of experts from different industries using IoTs, BDA, and CC deployment. Secondly, IoTs, BDA, and CC integration is still an emerging area and generalizing these findings may lead to insufficient conclusions. Additional studies should increase the sample size with IT security experts from different institutions whose responses to the digital security dimensions can be generalized. The findings of this study are limited to vulnerabilities and threats identified in the literature, hence responses from experts reflect the analysis of the results.

The integration of these digital technologies are still developing so future research should investigate and empirically validate the relationship and commonalities among the vulnerabilities and identified threats. Future studies should consider exploring other potential vulnerabilities and threats not mentioned in this study since digital security risks are multi-faceted. The data used for this study included perceptual views from security experts to generate research findings. Although perceptual data is encouraged in survey research, the use of operational data detailing vulnerabilities and threats from system logs, audit trails, and daily transactions or operational activities could further provide validity with digital security risk management. A combination of operational and perceptual data from audit trails or records from system logs could enhance future findings. Lastly, future research can look at other emerging vulnerabilities and threats emanating from the integration of BDA, blockchain technologies, and cloud computing.

# ACKNOWLEDGEMENTS

The authors wish to express their sincere appreciation to the heads of the banking units and the officials at the digital and security banking units for their voluntary participation in this study.

## Funding
The authors received no funding for this work.

## Competing Interests
The authors declare there are no competing interests.

## Author Contributions
- Abeeku Sam Edu conceived and designed the experiments, performed the experiments, analyzed the data, performed the computation work, prepared figures and/or tables, authored or reviewed drafts of the paper, and approved the final draft.
- Mary Agoyi conceived and designed the experiments, performed the experiments, analyzed the data, performed the computation work, authored or reviewed drafts of the paper, and approved the final draft.

- Divine Agozie conceived and designed the experiments, analyzed the data, prepared figures and/or tables, authored or reviewed drafts of the paper, and approved the final draft.

## Data Availability

The data are available in the Supplementary File.

## Supplemental Information

Supplemental information for this article can be found online at http://dx.doi.org/10.7717/peerj-cs.658#supplemental-information.

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
