# Peer review of "Digital security vulnerabilities and threats implications for financial institutions deploying digital technology platforms and application: FMEA and FTOPSIS analysis"

_PeerJ Computer Science, doi:10.7717/peerj-cs.658_

## Round 0.1 · original submission · Major Revisions

I have completed my evaluation of your manuscript. The reviewers recommend reconsideration of your manuscript following major revision. I invite you to resubmit your manuscript after addressing the comments below. The contribution should be discussed in detail.

·

Basic reporting

According to the authors, this study explores and provides a step-by-step analysis of the 19 potential vulnerabilities and threats failures affecting the integration of IoT. The work is somehow good and the problem researched by the author in the IoT domain is acceptable. Something I want to mention to improve the quality of the manuscript.

The problem should be better introduced in the abstract of the manuscript.

The research objectives and contributions are not clear enough as it mention in the introduction section line 107.

Experimental design

The methodological approach needs to be synthesized.

The presentation of the methodology section should be more strong.

No experimental design is mentioned.

The author mentioned in line no 367
"Based on the expert’s opinions derived through the questionnaire, the corresponding fuzzy numbers were calculated using the aggregated group matrix for Occ, Sev and Det for each vulnerability and threats identified. " which the method author used for the questionnaire.

The flow chart figure should be redrawn with high quality.

Validity of the findings

The finding is not clear in the manuscript. The author should add finding with bullets

Additional comments

Author work is better but some strong revision and manuscript should be revised thoroughly. some point I mentioned above for cosidration.

Reviewer 2 ·

Basic reporting

The authors used the failure mode effect analysis (FMEA) and fuzzy technique for order of preference by similarity for ideal solution (FTOPSIS) techniques to classify vulnerabilities and threats affecting IoT, big data, and cloud computing.

The structure of the paper includes the following sections: 1) introduction, 2) related work, 3) FMEA and fuzzy theory application, 4) methodology, 5) analysis and results, 6) discussion, and 7) conclusion.

The authors should define risk analysis in section 2.

The authors could include the following references:
N. Javadian, M. Kazemi, F. Khaksar-Haghani, M. Amiri-Aref and R. Kia, "A general fuzzy TOPSIS based on new fuzzy positive and negative ideal solution," 2009 IEEE International Conference on Industrial Engineering and Engineering Management, Hong Kong, 2009, pp. 2271-2274, doi: 10.1109/IEEM.2009.5373055.
S. Sun, Q. Wu and G. Liu, "Multi-level decision-making model for product design based on Fuzzy set theory," 2006 First International Symposium on Pervasive Computing and Applications, Urumqi, 2006, pp. 841-846, doi: 10.1109/SPCA.2006.297543.

Experimental design

The authors explain their methodology and provide a flow chart in figure 1. Their study requires a mixed methodology (qualitative and quantitative). The authors used the data of 234 questionnaires.

In Table 1, the authors define the failure modes and dimensions of IoT, BDA, and cloud computing. However, the failure modes appear more for a traditional infrastructure than for emerging technologies. Which is the explanation for this?

Validity of the findings

According to the results described by the authors, lack of back-up electric generator under the infrastructure risk dimension is the most dangerous threat. In your interpretation of results, why this threat is the most dangerous?

In the conclusions, the authors wrote "Fundamental limitation from this study is the reliance on security experts from financial institutions and their views may not entirely reflect views from other security experts from different industries with IoTs, BDA and CC deployment.", and I agree with this. From this point, I provide the following suggestions to the authors:
a) modify the title of the paper to "Digital Security Vulnerabilities and Threats implications on Financial Institutions using FMEA and FTOPSIS" or something similar.
b) change the idea of IoT, BDA, CC to security in the financial sector, providing evidence about the data breaches and cyber-attacks in the last year.
c) as future work, you can return to this paper including the opinion of security experts from IoT, BDA, and CC industries. In this way, the failure modes and dimensions will be more in line with emerging technologies.

Additional comments

The authors proposed an interesting technique to evaluate and classify threats and vulnerabilities; however, the dimensions and failure modes are not aligned to IoT deployment, Data analysis, and cloud computing technologies because each one has its vulnerabilities and threats. Moreover, security experts work in the financial area where the security requirements are very specific based on international standards (ISO 27000, PCI DSS, and others).

---

## Round 0.2 · Minor Revisions

According to the reviewers' comments, more details are needed.

·

Basic reporting

In this study, the author explores and provides a step-by-step analysis of the potential vulnerabilities and threats failures affecting the integration of IoT, Big Data Analytics, and Cloud Computing towards data management. The authors have done an extension towards the discussions of pervasive vulnerabilities and threats originating from integrating digital applications and platforms for data management. A detailed review and classification of these threats and vulnerabilities are 39 vital for sustaining the nature of digital integration for firms. The study highlights some major issues in the IoT Big data domain but there are some issues show be solved to improve the quality of work.

Experimental design

The methodology of the study is too much concise there should be some more details for clear understanding.

The methodology diagram should be redrawn and make some understandable especially the flow of data,

Validity of the findings

The results of this study are significant and also present in a well way.

Additional comments

As the author explores and provides a step-by-step analysis of the potential vulnerabilities and threats failures affecting the integration of IoT, Big Data Analytics, and Cloud Computing towards data management. The work is significant t but the grammar and typos errors are the weakness of the paper such as have seemingly --> has seemingly, from information ---> form an information system..

Reviewer 2 ·

Basic reporting

The authors improved the version of the paper, including the comments and suggestions by reviewers. The authors included the references suggested and changed the title of the article.

I suggest authors review the document to avoid finger mistakes. For example, on page 6, the authors wrote "resource u sage".

I suggest authors include a paragraph in the introduction section to provide a vision of financial institutions and the adoption of IoT, Big Data Analytics (BDA), and Cloud Computing.

Experimental design

The authors improved the version of the paper, including the comments and suggestions by reviewers. They provide a better explanation of their results.

Validity of the findings

The findings provide the first step to understand and resolve the problem of adopting emerging technologies by financial institutions.

Additional comments

I suggest authors include a paragraph in the introduction section to provide a vision of financial institutions and the adoption of IoT, Big Data Analytics (BDA), and Cloud Computing.

---

## Round 0.3 · Minor Revisions

The paper needs much more work especially in the part of the experimental results. The paper needs careful proofreading.

·

Basic reporting

In this study authors explore and provides a step-by-step analysis of the potential vulnerabilities and threats failures affecting the integration of IoTs, Big Data Analytics and Cloud Computing towards data management. A multi-dimensional analysis, Failure Mode Effect Analysis and Fuzzy Technique for Order of Preference by Similarity for Ideal Solution were combined to evaluate and rank the potential vulnerabilities and threats affecting emerging technology integration. A sample of 234 security experts from the banking industry with adequate knowledge in IoTs, Big Data Analytics and Cloud Computing were used for the survey. The results revealed a lack of back-up electric generators, a failure of firewall protection and a lack of information security audit as the high-ranking vulnerabilities and threats failures based on the closeness of coefficient scores to significantly affect Internet of Things, Big Data, and Cloud Computing integration. Further this study an extension of the discussions of pervasive vulnerabilities and threats originating from integrating digital applications and platforms for data management.

Experimental design

The experimental flow still needs improvement as I describe in the last review.
Explanation of methodology is not present in a well way.

Validity of the findings

Results and discussion are described in appropriate ways but I will suggest some graphical representation of results.

Additional comments

Typos and grammar still a problem in the article.

---

## Round 0.4 · Minor Revisions

We are happy to inform you that your manuscript will be accepted after the language check. Please use the benefits of fluent speaker.

·

Basic reporting

This study explores and provides a step-by-step analysis of the potential vulnerabilities and threats failures affecting the integration of IoTs, Big Data Analytics and Cloud Computing towards data management. A multi-dimensional analysis, Failure Mode Effect Analysis and Fuzzy Technique for Order of Preference by Similarity for Ideal Solution were combined to evaluate and rank the potential vulnerabilities and threats affecting emerging technology integration. A sample of 234 security experts from the banking industry with adequate knowledge in IoTs, Big Data Analytics and Cloud Computing were used for the survey. The results revealed lack of backup electric generators, a failure of firewall protection and a lack of information security audit as the high-ranking vulnerabilities and threats failures based on the closeness of coefficient scores to significantly affect Internet of Things, Big Data, and Cloud Computing integration.

Experimental design

No comments

Validity of the findings

No comments

Additional comments

The details are present in a well way as compared to the previous version of the manuscript. All concern point are discussed in detail. The author should still need to focus on grammar such as lots of mistakes of 's', 'es' are present in the manuscript.

---

## Round 0.5 · accepted · Accept

The authors have addressed all comments. It's our pleasure to inform you that your manuscript has been accepted for publication.

·

Basic reporting

The author addressed the challenges from an information systems perspective and has noted that more research is needed to identify potential vulnerabilities and threats affecting the integration of IoTs, BDA and CC for data management. They conducted a step-by-step analysis of the potential vulnerabilities and threats affecting the integration of IoTs, Big Data Analytics, and Cloud Computing for data management. They combined multi-dimensional analysis, Failure Mode Effect Analysis and Fuzzy Technique for Order of Preference by Similarity for Ideal Solution to evaluate and rank the potential vulnerabilities and threats.
The paper is now in good shape according to peerj acceptance rules.

Experimental design

No Comment.
Already improved by the author in the first revision.

Validity of the findings

Results are strong and present in well way

Additional comments

Please improve the grammar and typos before publishing.